# Electrochemical Affinity Biosensors Based on Selected Nanostructures for Food and Environmental Monitoring

**DOI:** 10.3390/s20185125

**Published:** 2020-09-08

**Authors:** Susana Campuzano, Paloma Yáñez-Sedeño, José M. Pingarrón

**Affiliations:** Departamento de Química Analítica, Facultad de CC. Químicas, Universidad Complutense de Madrid, E-28040 Madrid, Spain; pingarro@quim.ucm.es

**Keywords:** nanostructures, electrochemical biosensors, immunosensors, DNA sensors, food, environment

## Abstract

The excellent capabilities demonstrated over the last few years by electrochemical affinity biosensors should be largely attributed to their coupling with particular nanostructures including dendrimers, DNA-based nanoskeletons, molecular imprinted polymers, metal-organic frameworks, nanozymes and magnetic and mesoporous silica nanoparticles. This review article aims to give, by highlighting representative methods reported in the last 5 years, an updated and general overview of the main improvements that the use of such well-ordered nanomaterials as electrode modifiers or advanced labels confer to electrochemical affinity biosensors in terms of sensitivity, selectivity, stability, conductivity and biocompatibility focused on food and environmental applications, less covered in the literature than clinics. A wide variety of bioreceptors (antibodies, DNAs, aptamers, lectins, mast cells, DNAzymes), affinity reactions (single, sandwich, competitive and displacement) and detection strategies (label-free or label-based using mainly natural but also artificial enzymes), whose performance is substantially improved when used in conjunction with nanostructured systems, are critically discussed together with the great diversity of molecular targets that nanostructured affinity biosensors are able to quantify using quite simple protocols in a wide variety of matrices and with the sensitivity required by legislation. The large number of possibilities and the versatility of these approaches, the main challenges to face in order to achieve other pursued capabilities (development of antifouling, continuous operation, wash-, calibration- and reagents-free devices, regulatory or Association of Official Analytical Chemists, AOAC, approval) and decisive future actions to achieve the commercialization and acceptance of these devices in our daily routine are also noted at the end.

## 1. Introduction

Currently the food and environmental analytical fields demand accurate quantification of a wide variety of analytes such as food allergens, gluten, genetically modified organisms (GMOs), toxins, antibiotics, pesticide residues, bacteria, fungus, yeast, environmental oestrogens or heavy metals, in very diverse and complex matrices with the need for great sensitivity and selectivity and preferably using simple methods, with short turnaround times and applicable at the point of attention by any user.

To satisfy these demands, electrochemical affinity biosensors arise as an interesting alternative to conventional methodologies since they combine the specificity and efficiency of biological biorecognition processes with the advantages of electrochemical transduction in terms of rapidity, scalability and amenability for multiplexing analysis. When compared to other types of biosensors, such as those involving optical transduction, the implementation of electrochemical biosensors in electronic devices provides several advantages. Such advantages rely mainly to simplicity and affordability of the sensor setup in addition to high analytical accuracy for complex matrices where simple operation procedures and reduced detection times can be reached regardless of their turbidity or the presence of optically absorbing or fluorescing compounds. These features have moved electrochemical biosensors from promising candidates to essential basic tools to meet the demands of detection not only in many in situ and point-of-care (POC) circumstances for clinical diagnosis but also to achieve great development in other fields, such as food analysis, and environmental monitoring [1].

In this context, this review addresses the improved performance of electrochemical affinity biosensors when they incorporate single or hybrid nanostructures either as electrode modifiers or advanced labels (see Scheme 1). Importantly, the preparation of biosensing platforms involving nanomaterials can be properly modulated to enhance analytical characteristics such as sensitivity, selectivity, stability, conductivity, biocompatibility, and/or mimicked enzymatic activity. Under this common goal but following different strategies, nanostructured electrochemical biosensors for implementation of affinity reactions have been constructed with the specific purposes of increasing the loading of immunoreagents on the electrode surface or the amount of electroactive detectable species, and using the nanomaterial itself as the electrochemical signal-generating probe.

Taking into account this background, this review article provides a current and general overview of the outstanding properties offered by selected nanostructures-based electrochemical affinity biosensors involving dendrimers, DNA-based nanoskeletons, molecular imprinted polymers, metal-organic frameworks, nanozymes and magnetic and mesoporous silica nanoparticles for agri-food and environmental applications reported in the last 5 years. These nanostructures have been selected because they have been only recently used in electrochemical affinity biosensor configurations. In addition, the great versatility of modification and use they provide and the unique attributes they impart to the resulting biosensors allowing pioneering applications in the selected areas lead to predict an increasing interest for their use in the coming years.

After a brief introduction of the most relevant properties of the nanostructures, the opportunities and remarkable attributes of the resulting affinity biosensors are comprehensively discussed through selected representative methods classified according to the type of affinity biosensor and field of application to conclude with a section where main considerations, challenges and future perspectives are pointed out. 

## 2. Selected Nanostructures in Electrochemical Affinity Biosensing

Nanostructures have shown to play a key role in improving the performance of electrochemical affinity biosensors mainly in terms of sensitivity, selectivity and stability due to their small size, quantum size and interface effects [2,3,4,5,6,7]. Over the last years, nanomaterials have been widely exploited as electrode surface modifiers with three main purposes: (i) enhancing the conductivity of electrodes providing larger currents and higher signal-to-background ratios which result in a higher sensitivity; (ii) immobilizing (bio)molecules; and (iii) improving the biocompatibility of the electrode surface. Moreover, the use of nanomaterials as advanced labels exploit their catalytic and/or carrier properties to load large amounts of bioreagents, enzymes or redox mediators.

Here we briefly summarize the main characteristics and unique features imparted by selected well-known and less conventional nanostructures used in the preparation of electrochemical affinity biosensors for application in the past five years to the agro-food and environmental areas.

Gold nanoparticles (AuNPs) are one of the most commonly used nanomaterials in the preparation of nanostructures with a high versatility for application as electrode modifier, catalytic label, nanozyme, carrier of signal elements and electron transfer regulator in electrochemical affinity biosensing [8]. This is due to their special properties, including the large surface area, which enhance the amount of immobilized molecules also enabling their favorable orientation and spacing, as well as the outstanding conductivity, mimicked enzyme activity [9], chemical inertness and biocompatibility, the latter being crucial to preserve the bioactivity of attached bioreagents. AuNPs can be used in combination with carbon nanomaterials, such as reduced carbon oxide (rGO), carbon nanotubes (CNTs) or ordered-mesoporous carbon (OMC) for the preparation of nanostructured electrode surfaces applied to the construction of improved affinity biosensors.

Magnetic nanomaterials have demonstrated to be an efficient tool for electrochemical bioanalysis in complex matrices, which usually needs for an isolation of the target analyte, to minimize matrix effect and avoid the transducer surface fouling [10]. Iron oxide nanoparticles which exhibit superparamagnetic activity, have been prepared in sizes ranging from several units to tens of nanometers in diameter, and can be highlighted due to their large area, easy functionalization, peroxidase-like activity, improved assay kinetics and sensitivity, and simple handling by an external magnetic field [11,12,13,14]. However, these nanoparticles (MNPs) are more prone to agglomerate and to suffer losses during their more challenging handling than their micrometric analogues [15,16]. In electrochemical biosensing approaches, MNPs have been used mainly as nanosurfaces where affinity conjugation takes place without the need for applying costly and complicated modification procedures to the electrode platforms [17,18,19,20,21,22,23], but also as catalytic label and as nanocarriers of signalling molecules.

MNPs used more often in biosensing involve Fe_3_O_4_ (magnetite) and Fe_2_O_3_ (maghemite). This is due to the high biocompatibility and biodegradability that characterizes both materials [8]. A variety of uncoated MNPs are commercially available as well as MNPs modified with different coatings which minimize agglomeration, confer biocompatibility and allow modification with a wide variety of biomolecules, thus fueling the preparation of many biosensing designs. Among them, core-shell Fe_3_O_4_@Au [10,20,21,23] and Fe_3_O_4_@SiO_2_ [18,22] MNPs have been profusely utilized. In the first type, the presence of gold nanoparticles provides good conductivity, improves the adsorption capacity of biomolecules and makes it possible to use thiolated derivatives to incorporate functionalities [20], while SiO_2_ increases the stability of the nanoparticles providing a suitable surface for the binding of bioreactives [11]. Moreover, coatings of MNPs with polymers, such as polydopamine [17], allow increasing the immobilization capacity of biomolecules. 

As it is known, the so-called dendrimers are nanometric synthetic polymers, monodisperse, which possess a highly branched regular three-dimensional structure. These are widely employed as “soft” materials from recent times to custom design the chemical and physical properties of the electrochemical platforms and to amplify the signal in electrochemical affinity biosensors [24]. In this context, it is worth highlighting the special properties of these nanomaterials, especially their shape (ellipsoidal or globular), monodispersity, uniform structure with highly permeable internal cavities and the great capacity to carry functional moieties on their surface, which can be manipulated to design tailor-made and versatile dendrimers [25]. Due to their properties, dendrimers have also been used as particularly well suited nanomaterials for encapsulation of a variety of species and small particles through supramolecular hydrophobic or electrostatic interactions forming host-guest complexes. In particular, dendrimers with encapsulated metallic nanoparticles (mostly Au and Pt) display the advantages of both nanocomponents including high density of active groups for biomolecules immobilization, excellent structural homogeneity, remarkable conductivity, catalytic ability and good biocompatibility [26]. These hybrid nanomaterials have demonstrated to be particularly attractive options to be exploited in electrochemical affinity biosensing as both electrode modifiers [26,27,28] and efficient nanocarriers [29,30,31]. 

Over the past decade, nanozymes, i.e., artificial nanomaterials (metal oxide, metal, and carbon-based materials or their nanohybrids) with inherent activity similar to enzymes are gaining attraction as catalytic nanocarriers/labels [19,32,33] or electrode surface modifiers [34,35,36] in electrochemical affinity biosensing. Particularly, nanozymes with peroxidase-like activity are highly stable and low-cost alternatives to enzymes [9,37].

Mesoporous silica nanoparticles (MSNs) have aroused great interest in biosensing devices (mostly optical but also electrochemical) because they exhibit show good bioactivity and biocompatibility. In addition, their unique topology allows three distinct domains to be independently and straightforwardly functionalized: the silica framework, the uniform nanochannels/pores in the nanometer range, and the nanoparticle’s outermost surface [38,39,40,41,42]. MSNs, with versatile pore structure and functionality, can be fabricated with tunable size, shape and pore diameter using simple and quite affordable procedures, display excellent chemical and mechanical stability, high porosity (large surface areas and pore volumes) for high loading capacity, and are easily grafted with specific ligands [43]. MSNs can also be provided with stimulus-responsive gate-like molecular, supramolecular or bio-molecular ensembles at the external surface allowing on-command payload delivery from pores. Indeed, although they have been used more with optical detection, these gated MSNs-based nanodevices triggered by enzyme- or affinity reaction-mediated processes have been recently exploited in connection with electrochemical biosensing [44,45].

Apart from these artificial nanostructures, DNA has attracted rapidly increasing attention as a kind of nature-endowed functional nanomaterial. DNA molecules possess unparalleled self-recognition properties, offering great flexibility and convenience for the ‘‘bottom-up’’ construction of nanostructures with high controllability and precision [46,47]. In this context and in connection with electrochemical biosensors, the tetrahedral DNA nanostructures (TSPs), assembled on gold surfaces in a one-step, fast, easy and high performance process by means of the immobilization in a highly reproducible way of four especially designed single-stranded (ss)-DNA [46,48,49], lead to rigid, stable, and reproducible nanoscaffolds. 3D-DNA TSPs are suitable for immobilizing well oriented, separated in space detection probes, and at a remote distance from the electrode surfaces, in an environment that ensures improved efficiency in affinity reactions and minimization of non-specific adsorption without using additional backfilling reagents [50,51]. The TSPs interesting features, in terms of rigid structure, nanoscale addressability and versatile functionality, have led to their wide use in recent years in the development of electrochemical affinity biosensors using antibodies, aptamers and DNA oligonucleotides as recognition probes. The resulting biosensors exhibit attractive analytical characteristics for the determination of various types of molecular targets (DNAs, microRNAs or miRNAs, proteins, small molecules such as drugs or whole cells) in complex matrices. The properties of DNA molecules have also been exploited to form dendritic structure DNA (DSDNA) [52] and hemin/G-quadruplex nanowires [53] in amplification strategies for electrochemical affinity biosensors. It is worth to remark that TSPs have been used mostly as electrode modifiers [54] and DSDNA and hemin/G-quadruplex nanowires as labels [52,53].

Molecularly imprinted polymers (MIPs), synthesized from the appropriate monomer with the target compound as template, allow the construction of highly selective sensors, able to determine the target compound in the presence of other similar species in complex samples, by means of the cavities created to complement the target compounds by different interactions, shape or size characteristics. The MIPs low cost, reusability, and biomimetic properties compared to natural receptors are important properties for their use in biosensing [55]. By combining MIPs with nano-sized materials such as AuNPs, carbon nanotubes or graphene used as core materials during the synthesis of MIPs or as surface modifiers, leads to an enhancement of the conductivity, chemical stability, selectivity and surface area of the resulting scaffolds. In addition, sometimes electrocatalytic effects towards the electrochemical processes implied in the detection are achieved. Specific bio-modifiers (aptamers and DNAs) for the target compounds have been recently used in combination with MIPs nanostructures to increase their selectivity and sensitivity and develop affinity biosensors for pesticides [56,57].

Metal-organic frameworks (MOFs) are inorganic-organic hybrid materials composed periodically by metal ions and organic ligands [58]. The highly well-organized configuration, porosity and high surface area of MOFs are some of the advantageous properties that have driven their applications in different fields. Furthermore, they usually have intense electrocatalytic activity. As in other cases, the combination of MOFs with metallic nanoparticles or carbon nanomaterials can efficiently increase the number of catalytic active sites. Specifically, integration of AuNPs and MOFs also facilitates further modifications with biomolecules such as DNA, leading to the construction of sensitive biosensors [59,60].

## 3. Selected Nanostructures in Electrochemical Biosensing for Food Monitoring

In the field of food monitoring, mostly immunosensors and DNA sensors but also aptasensors and mast cells or lectins-based electrochemical biosensors have been developed, in connection with single or hybrid nanostructures used as electrode modifiers and, to a lesser extent, as nanocarriers and/or catalytic labels, for determining a great diversity of target analytes in a wide variety of food samples. Table 1 summarizes relevant features of the selected methods, comprehensively discussed in the following text, and classified according to the type of biosensor. Particular attention is given to the type of nanostructures and their role, target analyte, electrochemical technique and reported application. Although immunosensors and DNA sensors have been mostly prepared, aptasensors and others have been developed for the determination of: food allergens [18,61,62,63,64,65,66,67,68]; gluten [69,70]; GMOs [10,20,21,23]; toxins [32,71,72]; antibiotics [34,59,73,74]; pesticide residues [57]; bacteria [19,26,36]; fungus [22,75]; and yeast [76]. The biosensors were developed either in integrated formats or by coupling MNPs to conventional or screen-printed electrode substrates.

The development of integrated biosensors implies the use as scaffolds of conventional (glassy carbon, GCE, and gold, AuE) or screen-printed (SPCE, SPGE) electrodes, in many cases modified with single (AuNPs, CNTs, CAs, CNFs, G, GO, GQDs) or hybrids (PAMAM(Au), AuNPs-rGO, chitosan-modified multi-walled carbon nanotubes, CS-MWCNTs) nanomaterials, as well as with diverse nanostructures of MIPs and MOFs. A limited number of biosensors uses unmodified electrode substrates and nanomaterials as nanocarriers and/or catalytic labels [32,71,74]. Some methods combine the use of different nanomaterials with different roles within the same design. In addition, Fe_3_O_4_@Au MNPs have been employed as solid supports for the preparation of immune-, DNA-, mast cells- and lectins-based biosensors.

The constructed biosensors involve single, sandwich, competitive and displacement affinity reactions with label-free or label-based detection strategies. The label-based strategies use mainly natural (alkaline phosphatase, AP, and horseradish peroxidase, HRP) or artificial enzymes (nanozymes mostly with peroxidase-like activity) together with the appropriate enzyme substrate systems (3-IP/Ag^+^, HQDP, Aniline/H_2_O_2_, H_2_O_2_, HQ/H_2_O_2_, TMB/H_2_O_2_ and Thi/H_2_O_2_). Electrochemical detection techniques include both time-dependent, such as voltammetries (CV, SWV and DPV) and amperometry, and frequency-dependent (EIS) sensing techniques and even field-effect transistor (FET) transduction. The developed biosensors exhibit selectivity and sensitivity for the target analytes with detection limits (LODs) in the range of ng mL^−1^ for proteins, aM-pM for genetic analytes, and less than 100 colony forming units (cfus) mL^−1^ for bacteria, yeast and fungus cells. Quantification of target analytes are reported for a wide variety of food samples: cookies, chocolate, crackers, biscuits, durum wheat pasta, breadcrumb, peanut, pork, crucian carp, brown shrimp, manioc, rice, corn, flour, honey, fruits, soybean seeds, cat feed, yogurt, milk and wine.

Some illustrative examples of the methods summarized in Table 1 are discussed below. 

Commercial SPCEs modified with AuNPs [69] or modified in the laboratory by electrodeposition [62,63,64,68] have been used for the development of label-free [68] or enzyme-labelled [62,63,64,69] immunosensors for the determination of protein allergens in different food matrices. FET [65] and chronoamperometric [70] immunosensors for Ara h 1 or gliadin have been constructed using SPCEs modified with SWCNTs or CNFs, respectively. Graphene (G) and spongy gold film-modified CS-MWCNTs have been used to modified SPCE and GCE to develop very sensitive aptasensors and hairpin-based DNA sensors for targeting β-LG (LOD = 20 pg mL^−1^ [67]) and a Ara h 1 characteristic DNA fragment (LOD = 13 aM [61]). An aptasensor has been proposed also for OTA determination in corn samples involving a displacement assay and the use of carbon aerogels (CAs) as nanocarriers of a complementary DNA (cDNA) sequence [71]. Zhang et al. [26] reported a sandwich immunosensor for the sensitive determination of *E. coli* by combining the advantages of a nanocomposite prepared by encapsulating AuNPs in a PAMAM dendrimer as electrode modifier. This nanocomposite allowed increasing the immobilized capture antibody (cAb) loading as well as enhancing the electron transfer process. In addition, an amplification factor occurred by using carbon nanotubes as nanocarriers of multiple DAb and HRP molecules (DAb-CNT-HRP) (Figure 1). By measuring the DPV oxidation peak of polyaniline (PAn) catalytically deposited by DAb-CNT-HRP nanoprobes, the method achieved a LOD of 50 cfu mL^−1^(S/N = 3), within 3 h.

Another interesting strategy is that proposed by Villalonga’s group for the determination of contaminating yeasts and fungi using disposable electrodes nanostructured with nanocomposites of reduced graphene oxide and AuNPs (AuNPs-rGO, Figure 2) [75] or graphene oxide (GO) [76]. In both methods, covalent immobilization of the specific capture antibody using carbodiimide/ succinimide chemistry on the carboxylic groups of the surface nanostructured with 3-mercaptopropionic acid (MPA) was performed. The captured yeast or fungus were enzymatically labelled with HRP-conjugated Concanavalin A lectin (HRP-ConA), able to specifically recognize the carbohydrate moieties in the mannoproteins at the surface of the yeast and/or fungus cells. The amperometric measurements were made using the H_2_O_2_/hydroquinone (HQ) system. The immunosensors were employed for have demonstrated applicability for the analysis of contaminated red and white wine samples with LODs of 56 and 6 cfu mL^−1^ for *Brettanomyces bruxellensis* and *Saccharomyces cerevisiae*, respectively.

Khan et al. [66] reported an aptasensor fabricated through a low-cost inkjet-printing method, which enabled both aptamer density control and high-resolution patternability (there is no discussion about the orientation of the aptamer) for the impedimetric determination of lysozyme. The method profited the strong affinity between CNT and the single-stranded DNA to immobilize the aptamers on the working electrode by printing the ink containing the dispersion of the CNT-aptamer complex. Figure 3a, shows as in the absence of target analyte, the electron transfer from the redox probe [Fe(CN)_6_]^4−/3−^ to the electrode was hindered due to the negatively charged backbone and the insulating nature of the aptamers. This behaviour provoked a big charge transfer resistance, R_ct_, corresponding to the semicircle diameter in the Nyquist plot (Figure 3b). In its presence of the analyte, the aptamer unwraped itself from the CNT due to the preferential binding to the lysozyme (Figure 3c) thus opening up the path for electrons to flow from the redox probes to the working electrode (small R_ct_ in the Nyquist curve, Figure 3d). This inkjet printing-based aptasensor achieved an LOD of 90 ng mL^−1^, good selectivity and reasonable shelf-life (~21 days at room temperature), thus appearing as a good alternative for point-of-care diagnostics by enabling low-cost, label-free, fast detection, and on-demand printability.

Other attractive methods exploited the catalytic properties of particular nanomaterials, used as electrode modifiers [34,36] or nanocarriers [19,32] to mimic peroxidase activity. AuNPs and GQDs were employed as AuE modifiers to develop direct aptasensors and immunosensors for the determination of kanamycin (KANA) [34] and *Yersinia enterocolitica* [36] in milk and honey, respectively. The aptasensor used the increased DPV signal measured in the presence of Thi/H_2_O_2_ due to the peroxidase activity recovery of AuNPs by the displacement of the aptamer adsorbed on their surface and corresponding to the reduction of the oxidized Thi generated in the reduction of H_2_O_2_ by AuNPs. In the case of the immunosensor, the chronoamperometric response measured in the presence of H_2_O_2_ decreased in the presence of the target analyte due to the inhibited electron transfer of the laminated GQDs on the Au surface. Moreover, the artificial peroxidase properties of nanocomposites of 5,10,15,20-Tetraphenyl-21H,23H-porphine cobalt flat stacking on the reduced graphene oxide with platinum nanoparticles (PtNPs/CoTPP/rGO) [32] and Au@Pt nanoparticles onto neutral red (NR) functionalized rGO (rGO-NR-Au@Pt) [19] used as nanocarries of DAb or DAb+HRP molecules, were exploited in indirect competitive and sandwich immunosensors for determining AFB_1_ in peanut and *E. coli* O157:H7 in pork and milk samples, respectively. In these methods, DPV and CV techniques were used for the electrochemical measurement of H_2_O_2_ using TMB [32] or Thi [19] as redox mediators, respectively. 

The use of MNPs as solid supports to develop DNA-, mast cell- and lectin-based bioassays for the determination of GMOs, protein allergens and yeast cells deserves to be mentioned. 

An illustrative example is the use of Fe_3_O_4_@Au MNPs modified with self-assembled monolayers (SAMs) of a mercaptoacid further modified covalently using carbodiimide/succinimide with aminated DNA capture probes to develop DNA sandwich hybridization strategies involving FITC or DIG modified-detector probes enzymatically labelled with HRP-conjugated Fab fragments [10,20,21,23]. By using amperometry in the presence of H_2_O_2_/HQ upon magnetic capturing of the modified MNPs onto SP(d)CEs or homemade AuEs, these methods exhibited a high sensitivity (pM-nM) for the determination of synthetic DNA fragments characteristic of transgenic maize and soybean. The biosensors were applied to determine the PCR amplicons obtained from certified samples, soybean seeds, cat feed and maize flour.

Jiang et al. [18] reported an electrochemical mast cell sensor based on the use of cationic magnetic fluorescent nanoparticles (CMFNPs) prepared by encapsulating Fe_3_O_4_@SiO_2_@FITC with lipidosome to transfect into RBL-2H3 cells activated by an allergen antigen. The R_ct_ values measured by EIS in the presence of [Fe(CN)_6_]^4−/3−^ upon magnetic capturing the CMFNP-transfected RBL-2H3 cells onto a GCE decreased in a dose-dependent manner with the concentration of the antigen used to activate the mast cells. The sensor provided LOD values of 0.03 μg mL^−1^ and 0.16 ng mL^−1^ for shrimp (tropomyosin, Pen a 1) and fish (parvalbumin, PV) allergens determination, respectively. 

More recently, Fe_3_O_4_@SiO_2_ MNPs modified with Con A or a specific antibody as nanocaptors have been employed to develop electrochemical affinity bioassays for the determination of total yeast or Brett cells in wine, respectively [22]. In both cases, the captured cell was enzymatically labelled with ConA-HRP. Amperometric transduction was used upon capturing the magnetic nanobioconjugates at SPCEs. LOD values of 8 and 5 cfu mL^−1^ were obtained for *Brett* and total yeast cells in spiked red wine samples. Zhu et al. [19] exploited also Fe_3_O_4_@SiO_2_ MNPs as immunocaptors to develop a sandwich configuration for determining *E. coli* O157:H7 using voltammetry. 

As it has been already mentioned, MIPs constitute an excellent alternative for the preparation of sensing surfaces for recognizing specific target compounds [77]. Recently, the integration of aptamers with these structures has made possible the construction of hybrid biosensors with better affinity and selectivity features. In addition, the use of nanomaterials expands monitoring capabilities by increasing the number of affinity positions, which improve the detection properties of MIPs. Roushani et al. [73] reported an interesting aptamer-MIP hybrid configuration involving 3-aminomethyl pyridine - functionalized graphene oxide (3-ampy-rGO) and AgNPs on a GCE for the EIS determination of chloramphenicol (CAP) in milk with a LOD as low as 0.3 pM. Once the CAP complex-amino-aptamer (NH_2_-Apt[CAP]) was attached to the AgNP/3-ampy-rGO/GCE by Ag-N bonding, the bio-nanostructure was stabilized by electropolymerization of resorcinol. The same group developed an aptamer-MIP hybrid composite using gold nanorods as the appropriate matrix for the covalent immobilization of the aptamer-MIP, which was used for the detection of chlorpyrifos (CPS) in apples and lettuce [57]. Interestingly, the authors selected *o*-phenylenediamine (*o*-PD) and *o*-dihydroxybenzene (*o*-DB) as the functional monomers to form –NH and –OH groups enhancing recognition sites and taking advantage of the good chemical and mechanical stability and high electron transfer efficiency. This strategy combining aptasensing and molecular imprinting allowed designing a highly sensitive method. A similar MIP/DNA biosensor was reported for the determination of bisphenol A (BPA) in milk and water samples [78]. In this design, AuNPs were electrodeposited onto a GCE and a mixture of thiolated DNA sequence (p-63) and the template were incorporated followed by electropolymerization of pyrrole. The impedimetric measurements provided a linear range extending from 0.5 to 5000 fmol L^−1^ with a LOD value of 80 amol L^−1^. 

The fascinating properties of MOFs such as high pore volume, large surface area, tunable structures, and open metal sites, allow the combination of nanomaterials-based MOFs with biomolecules, especially aptamers, to obtain better affinity and enhanced responses. A variety of aptasensor-MOF hybrids have been prepared recently for application in food analysis, with drug residues, especially antibiotics, as the most usual target compounds. An illustrative example is the work reported by Meng et al. [59] for the determination of streptomycin (STR) in milk using a MOF-based bio-bar code and enzyme-assisted target recycling for signal amplification. Figure 4 illustrates schematically the preparation process by using AuNPs modified-SPCEs where a mixed monolayer of thiolated cDNA/aptamer duplexes (dsDNA) and 6-mercapto-1-hexanol (MCH) was immobilized. In the presence of target STR, the aptamers from dsDNA were removed assisted by Exo I enzyme, and Ru(NH_3_)_6_^3+^ was then surface confined providing the electrochemical response.

In an interesting application, He and Dong [79] used aminated HP-UiO-66, a hierarchically porous (HP) nanoscale MOF based on Zr (IV) as signal tag by aptamer binding and encapsulated methylene blue (MB) mediator. Complementary ssDNA was attached by Au-S bond to a gold electrode modified with AuNPs/chitosan/ZnO nanoflowers, which provided signal amplification due to the large specific surface area and the high number of signal tags that can be loaded. The resulting aptasensor-MOF hybrid was applied to the detection of patulin (PAT, 4-hydroxy- 4H-furo[3, 2-c]pyran-2(6H)-one), a cytotoxic product of *Penicillum*, *Aspergillus* and *Byssochlamys* funguses metabolism process widely present in decaying fruits. In the presence of PAT, the aptamer-cDNA dissociated releasing some tags, which resulted in the decrease of the electrochemical responses. The change in DPV current was proportional to the PAT concentration from 5 × 10^−8^ to 0.5 μg mL^−1^, allowing the determination of PAT in apple juice samples.

## 4. Selected Nanostructures in Electrochemical Biosensing for Environmental Monitoring

Regarding environmental monitoring, immunosensors exploiting single or hybrid nanomaterials have been reported for bacteria, their toxins or oestrogens. Aptasensors and DNA sensors involving selective aptamers, hairpin probes and catalytic DNAzymes together with different nanostructures employed as electrode modifiers and nanocarriers of signalling elements have been focused mainly to the determination of heavy metals. Table 2 compares the main characteristics of representative electrochemical affinity biosensors for environmental monitoring, grouping the biosensors again according to the bioreceptor type.

Immunosensors have been developed mainly using enzyme-free, integrated formats with conventional electrodes (GCE and ITO) modified with a single nanomaterial (polyacrylonitrile nanofibers, PANnf’s, CNTs, CNFs, ZnO NPs) or nanohybrids (PPI-AuNP and CoFe_2_O_4_/rGO). Nevertheless, electrochemical immunoassays involving sandwich configurations and enzymatic amplification on the surface of Fe_3_O_4_@pDA MNPs can be also mentioned. It is important to note that some of these approaches exploit the mimicked HRP activity imparted by some nanomaterials (Au@Pd NRs) to prepare label-free immunosensors. Similarly to food analysis, hybrid MOF-based biosensors are gaining more and more interest. An immunosensing representative example is the configuration reported by Gupta et al. [80] for the determination of *E. coli* in spiked lake water. In this work, a MOF-based electrochemically active platform involving Cu_3_(BTC)_2_, where BTC is 1,3,5- benzenetricarboxylic acid, and PANI was prepared and bio-interfaced with anti-*E. coli* antibodies onto an indium-tin-oxide (ITO) electrode. Impedimetric measurements with the resulting biosensor allowed detecting as low as 2 cfu mL^−1^
*E.coli* in a short response time of around 2 min.

Aptasensors and DNA sensors have been used mainly to the determination of heavy metals such as Pb^2+^ and Hg^2+^ developing strategies that combine the selectivity of aptamers, hairpin probes and catalytic DNAzymes for the metal to be determined with the advantageous properties of different nanostructures used as electrode modifiers (ERGO, ordered mesoporous carbon–gold nanoparticle, OMC–GNPs, AuNPs and DNA tetrahedral nanostructures, TSPs), artificial enzymatic nanolabels ((Fe−P)_n_-MOF) and nanocarriers of signaling elements (Pt@PdNCs). Other reported original approaches involve the use of multifunctional hemin/G-quadruplex nanowires to serve simultaneously as bienzyme and direct electron mediator or MSNs as molecular gates. All these methods allow sensitive (fM-pM) and selective determination of the target metal mostly in spiked environmental soil and water (river, lake, tap, pool, valley and secondary treated waste) samples. Some selected examples are discussed below.

Martin et al. [17] reported a sandwich immunosensor for the determination of *Legionella pneumophila* (*L. pneumophila*) using Fe_3_O_4_@pDA MNPs as solid support for the bioassay and amperometric detection at SPCEs. The method provided selective and quite sensitive determination (LOD values of 10^4^ cfu mL^−1^ and 10 cfu mL^−1^, with a preconcentration step) and showed applicability to perform the determination in inoculated water samples.

Zhang et al. [35] developed a competitive immunosensor for estradiol by taking advantage of the high electrical conductivity, open porous structure and large loading capacities of CoFe_2_O_4_/reduced graphene oxide nanohybrid and the HRP-mimicked activity of Au@Pd nanorods. The immunosensor achieved a LOD of 3.3 pg mL^−1^ and was used to analyze natural water with satisfactory results.

Label-free and single-immunoreaction biosensors have been reported using electrode substrates nanostructured with PPI-AuNP nanocomposite [81], PANnf’s [82], ZnO NPs [83], CNTs [84] and CNFs [85] for *Vibrio cholerae* [82] and its characteristic toxins [81,82,85] or antibodies [84]. Although all these immunosensors exhibit good analytical characteristics in terms of sensitivity and selectivity, none has been applied to the analysis of real samples. As an example, Figure 5 displays the Nyquist plots obtained in the presence of [Fe(CN)_6_]^4−/3−^ at an immunosensor developed for anti-cholera toxin antibodies detection prepared by coordinative binding of the biotinylated cholera toxin on a GCE modified with MWCNTs and poly(pyrrole-NTA)/Cu^2+^ by electropolymerization.

Molecular recognition processes combined with aptamers are the basis of biosensors prepared for the specific detection of 2,4,6-trinitrotoluene [86] and urea [87]. These methods combine in a synergistic way the recognition properties of aptamers and molecular imprint sites providing excellent specificity and sensitivity. Urea is extensively used as fertilizer in agriculture and it can contaminate ground water and aquatic systems. A MIP-aptamer nanohybrid involving AuNPs and carbon nanotubes was prepared and immobilized onto a GCE for the determination of urea in environmental samples. When DNA aptamers and MIP were impregnated with the target molecule, the cavities were blocked, and the impedimetric response decreased [87]. In the case of the determination of TNT, an amino-aptamer with inherent affinity for the nitro-derivative explosive was mixed with the target compound prior to covalent immobilization onto a GCE modified with AuNPs@fullerene (C_60_). For the MIP fabrication, dopamine was electropolymerized around the aptamer/TNT complex, and the resulting nanohybrid receptor (aptamer-MIP) merged the recognition properties of the single receptors [86].

An hybrid aptasensor for the sensitive determination of heavy metal ions was prepared with a Fe(III)-based MOF-derived core–shell nanostructure involving mesoporous Fe_3_O_4_@C nanocapsules. The large specific surface area of the nanostructure deposited onto a gold electrode allowed a strong bio-binding with aptamer strands. The aptasensor was applied to the determination of Pb^2+^ and As^3+^ by combining the conformational transition interaction caused by the formation of the G-quadruplex between a single-stranded aptamer and the adsorbed metal ions. This strategy provided LOD values of 2.27 and 6.73 pM for Pb^2+^ and As^3+^, respectively [88].

Regarding nucleic acid-based biosensors, it is remarkable the simple preparation and operation of the aptasensor reported by Yu et al. [89] for the label-free determination of Pb^2+^ by attaching via Π-Π interaction a guanine-rich DNA aptamer tagged with methylene blue (MB) to the surface of GCE modified with ERGO (Figure 6). In the presence of the target ion the aptamer was folded to a G-quadruplex structure and detached from the ERGO/GCE, producing a decrease in the reduction peak current of the MB tag recorded by CV and DPV. This aptasensor showed remarkable features in terms of selectivity, sensitivity (LOD of 0.51 fM) and reusability by simply DOTA (1,4,7,10-tetraazacyclododecane-1,4,7,10-tetraacetic acid) addition to the analytical solution.

MIPs prepared by monomer polymerization in the presence of both the template molecule and DNA lead to enhanced sensitivity and specificity of the detection. An illustrative example is the fabrication of a DNA biosensor for 2,4-dichlorophenoxyacetic acid (2,4-D) where a MIP layer was prepared with poly ortho-phenylenediamine (PoPD) together with entrapped DNA on a pencil graphite electrode (PGE) modified with a mixture of chitosan and MWCNTs [56]. A linear range extending between 0.01 and 10 pM and a LOD value of 4.0 fM were achieved.

It is worth to remark the personal glucometer (PGM) biosensor developed by Liang et al. [44] for quantification of Hg^2+^ based on the target-responsive release of glucose from single-strand wrapping DNA sealed MSNs (DNA-gated MSNs). Figure 7 shows as, upon the addition of Hg^2+^ and an assistant DNA, the T-Hg^2+^-T base pairing can detach wrapping DNA from MSNs and induced the formation of wrapping and assistant DNA duplex. The formed duplex was then recognized by Exo III, which can digest the wrapping DNA from 3′-hydroxyl terminus, thus releasing Hg^2+^ and leading to the continuous detaching of wrapping DNA from MSNs. As a result, the “gate” was opened, and allowed the pore-trapped glucose diffuse out of MSNs for PGM readout. The value of the amperometric signal provided by the PGM was proportional to the Hg^2+^ concentration. This method provided a LOD of 0.1 nM, far below the maximum permissible content of Hg^2+^ established by the United States Environmental Protection Agency (EPA) in drinking water (10 nM).

DNAzymes have also been used in combination with nanomaterials and other attractive amplification strategies in the development of electrochemical biosensors for the determination of heavy metals.

For instance, the selective cleavage of catalytic DNAzymes in the presence of Pb^2+^ was exploited for the preparation of an impedimetric DNA sensor [90]. The authors used a GCE modified with OMC–AuNPs through L-lysine and further with an electrodeposited AuNPs film, as a scaffold to immobilize a thiolated DNA probe able to hybridize with DNAzyme catalytic beacons. As it is shown in Figure 8, in the presence of Pb^2+^, the DNAzyme cleaved the substrate strand into two DNA fragments and the R_ct_ measured by EIS in the presence of [Fe(CN)_6_]^3−/4−^ decreased linearly with the Pb^2+^ concentration between 5 × 10^−10^ and 5 × 10^−5^ M.

Electrochemical biosensing of Pb^2+^ was also made by coupling the high specificity of DNAzymes, an efficient amplification strategy using catalytic hairpin assembly (CHA) induced by strand replacement reaction, the in situ layer-by-layer assembly of dendritic structure DNA (DSDNA) labelled on Pt@Pd nanocages (Pt@PdNCS) with large surface area and excellent catalytic performance, and the mimicking peroxidase activity of manganese(III) meso-tetrakis (4-N-methylpyridiniumyl)-porphyrin (MnTMPyP) molecules embedded into the formed DSDNA scaffold [52]. As it is schematically displayed in Figure 9, the presence of Pb^2+^ provoked that the substrate strand (S1) of the Pb^2+^-specific DNAzymes was specifically cleaved into two fragments, one of which (rS1) hybridized with the hairpin DNA (H1) attached to the AuNPs-GCE and was further replaced by another hairpin DNA (H2) by the hybridization reaction of H1 with catalytically recycled H2. Finally, Tb-S3-Pt@PdNCs and Tb-S4-Pt@PdNCs, which are Pt@PdNCs bioconjugates labelled with DNA S3 or S4 and with toluidine blue (Tb) as the electroactive marker, were captured through the hybridization of S3 and H2, S3 and S4, onto the resultant electrode surface leading to the formation of DSDNA triggered by layer-by-layer assembly which greatly facilitated the MnTMPyP immobilization. Using DPV in the presence of H_2_O_2_, this method achieved a LOD of 0.033 pM together with good performance in the analysis of tap and lake water samples using the standard addition method.

Wang et al. [54] reported a label-free Pb^2+^ biosensor involving the immobilization of a specific DNAzyme (composed of a DNAzyme strand and a substrate strand) on a TSP assembled to an AuE. The TSP allowed a better control of the density and orientation of the probe thus improving the DNAzyme reaction efficiency. In the presence of Pb^2+^ the substrate strand is cleaved and released a “G-rich” oligo able to form a G-quadruplex/hemin complex which generated a detectable signal by CV in the presence of H_2_O_2_ (Figure 10). This biosensor displayed a LOD of 0.008 nM, which is 9000 times lower than the safety limit of EPA (72 nM).

Another electrochemical biosensor for Pb^2+^ was prepared by exploiting the use of multifunctional hemin/G-quadruplex nanowires, formed by rolling circle amplification (RCA), both as bienzyme and direct electron mediator [53]. Figure 11 shows as the Pb^2+^-dependent DNAzyme was cleaved specifically by the target Pb^2+^ allowing primers capture which next triggered RCA producing amounts of long hemin/G-quadruplex nanowires in the presence of hemin. When nicotinamide adenine dinucleotide (NADH) was present, the formed hemin/G-quadruplex nanowires acted as NADH oxidase and HRP-mimicking DNAzyme in a synchronized way to accelerate direct electron transfer (DET) from hemin to the electrode surface. This electrochemical reaction resulted in a DPV signal proportional to the Pb^2+^ concentration in the 0 fM to 200 nM concentration range. This biosensor achieved a LOD of 3.3 fM and was used for the determination of Pb^2+^ in tap water using the standard addition method.

Cui et al. [33] reported the use of an hairpin DNA-modified SPCE and DNA functionalized iron−porphyrinic metal−organic framework (GR−5/(Fe−P)_n_-MOF) probes which combine the specific cleavage of GR−5 in the presence of Pb^2+^ with the excellent mimic peroxidase ability of (Fe−P)_n_-MOF. In the presence of Pb^2+^, the short (Fe−P)_n_-MOF-linked oligonucleotide fragment, produced by specific cleave of GR−5 at the ribonucleotide (rA) site, hybridized with the surface tethered hairpin DNA. The chrono-amperometric response for Pb^2+^ provided by the biosensor using the H_2_O_2_/TMB system was enough selective and sensitive (down to 0.034 nM) to allow the accurate analysis of spiked soils. Similarly, a scheme for amplification was reported involving a target-triggered nuclear acid cleavage of Pb^2+^-specific DNAzyme combined with Fe-MOF modified with PdPt NPs as signal tags [91] (Figure 12). DNAzyme was immobilized onto a streptavidin-reduced graphene oxide (rGO) and tetraethylene pentamine (TEPA) - AuNPs modified gold electrode where a new single DNA was produced in the presence of Pb^2+^ from the DNA strand on the interface. Finally, Fe-MOFs/PdPt NPs modified with a hairpin DNA able to hybridize with the substrate strand of the DNAzyme attached to the electrode was used for signal amplification catalyzing the H_2_O_2_ electrochemical current. A linear relationship between such responses and Pb^2+^ concentrations ranging from 0.005 to 1000 nM with a LOD value of 2 pM was obtained.

It is worth noting here that, very recently, electrochemical affinity biosensors using other less conventional nanostructures than those we have focused in this work have been prepared to provide other important features beyond sensitivity and selectivity, such as the possibility of self-calibration. As an example, Lee et al. [92] designed a self-calibrating multiple electrochemical aptasensor for the detection of avian influenza viruses (AIV) which involved dual rod electrodes modified with the specific anti-AIV NP aptamer (AptAIV) and Aptcon as the control aptamer supported onto a 3D porous silica nanostructure for targeting an AIV nucleoprotein (NP). The sensor platform consisted of an anti-AIV-NP-aptamer-capped and MB loaded electrode (AptAIV-MB@electrode), together with a control-aptamer-capped electrode (Aptcon-MB@electrode). The control aptamer had a length equivalent to the specific aptamer (anti-AIV-NP aptamer) but the random sequence did not specifically bind with AIV NPs thus providing self-correction for a drifted baseline signal. As Figure 13 displays, the binding interaction between AptAIV and AIV NPs triggers the detachment of AptAIV from the outer surface of the AptAIV-MB@electrode. This leads to uncapping of the porous film structure and MB releasing, thereby decreasing the MB electrochemical current measured by CV. The method achieved a LOD value in the nanomolar-range, and the determination was reliable under various pH and ion strength conditions and even in the presence of cell lysis debris. Furthermore, the biosensor was preliminarily applied to the analysis of negative oral and cloacal swab samples collected from chickens experimentally infected with H9N2 viruses.

## 5. General Considerations, Challenges and Perspectives

Undoubtedly, the use of nanostructures as electrode modifiers and/or specialized labels has been responsible for the excellent capabilities that electrochemical affinity biosensors exhibit in terms of sensitivity, selectivity and stability and the booming of these biosensors development and exploration in many application fields during the last years.

This review article shows an overview of the versatility and possibilities provided by electrochemical affinity biosensors for applications in the agro-food and environmental fields, which are less covered in the literature than the clinical field, by discussing in a comprehensive way the relevant aspects of representative methods reported in last 5 years literature. Innovations in the manufacture, modification and use of particular nanostructures are largely responsible of the attractive features and applications showed by the selected biosensors. With the aim of offering a newer and more updated point of view, special attention is paid to biosensors that employ less conventional nanostructures such as MNPs, dendrimers, nanozymes, MSNs, TSPs, MIPs, MOFs and their nanohybrids.

In these biosensors, nanomaterials have been used as modifiers or advanced labels to improve the conductivity and the biocompatibility of the electrode surface and to exploit their catalytic, mimicked enzymatic activity and/or carrier properties to load large amounts of (bio)molecules.

Particularly noteworthy are the versatility, catalytic ability and biocompatibility of AuNPs, the use of MNPs as efficient solid nanosupports for bioassays allowing straightforward determinations directly in complex matrices, dendrimers as “soft” nanomaterials to encapsulate nanoparticles and tailor the chemical and physical electrode surface properties, nanozymes as highly stable and low-cost alternatives to enzymes, the versatile pore structure and functionality of MSNs to be used as molecular gates, and the features imparted by hybrids of aptamer/DNA and MIPs or MOFs, both as electrode modifiers and signalling tags.

Applications in the food field imply mostly immunosensors and DNA sensors but also aptasensors and others involving mast cells and lectins. The biosensors are prepared through both integrated and MNPs-based formats and target food allergens, gluten, GMOs, toxins, antibiotics, pesticide residues, bacteria, fungus and yeast cells, have been developed.

In the integrated biosensor configurations, the nanomaterials (AuNPs, CNTs, CAs, CNFs, G, GO, GQDs) or their hybrids (PAMAM(Au), AuNPs-rGO, CS-MWCNTs, MIPs and MOFs) are used mainly as electrode modifiers and less explored as nanocarriers and/or catalytic labels. Some designs combine the use of different nanomaterials with different roles. Fe_3_O_4_@Au MNPs have been used as solid nanosupports for the construction of immune-, DNA-, mast cells- and lectins-based biosensors.

Electrochemical affinity biosensors, involving single, sandwich, competitive and displacement affinity reactions and label-free or label-based detection strategies (using mainly natural but also artificial enzymes), allow the sensitive and selective determinations of the target analytes and their determination in a wide variety of food samples through quite simple and straightforward protocols. However, their applicability for the detection of bacteria and bacterial toxins has been little evaluated.

Nanocomposites prepared by encapsulating AuNPs in PAMAM, which allows both to increase the immobilized capture antibody loading and accelerate the electron transfer process, have been used in the preparation of immunosensors for the sensitive determination of bacteria. Aptasensors prepared by a low-cost inkjet-printing method of a CNT-aptamer complex dispersion have been constructed for the impedimetric determination of lysozyme. “Off-on” aptasensors and immunosensors for the sensitive determination of KANA and *Y. enterocolitica* have been developed by exploiting the HRP-mimicked activity of AuNPs.

Fe_3_O_4_@Au MNPs and Fe_3_O_4_@SiO_2_ MNPs have been used as solid supports to develop DNA, mast cell- and lectin-based bioassays for the determination of GMOs, protein allergens and yeast cells. Fe_3_O_4_@SiO_2_@FITC were used for the determination of shrimp and fish allergens once encapsulated with lipidosome and transfected into activated RBL-2H3 cells or modified with Con A/specific antibody as efficient nanocaptors for total yeast, *Brett* and *E. coli* O157:H7 cells.

A wide variety of aptamer/DNA and MIPs or MOFs nanohybrids have been exploited in electrochemical biosensors for the determination of antibiotics, insecticides, BPA, and cytotoxic products in a wide variety of food samples.

Regarding the environmental field, single nanomaterials (PANnf’s, CNTs, CNFs, ZnO NPs) or their nanohybrids (PPI-AuNP, CoFe_2_O_4_/rGO and Cu_3_(BTC)_2/_ PANI/MOF) have been exploited in immunosensors for bacteria, bacterial toxins and environmental oestrogens.

Aptasensors and DNA sensors have been applied mainly to the determination of heavy metals through the combination of the selectivity of aptamers, hairpin probes and catalytic DNAzymes with different nanostructures used as electrode modifiers (ERGO, ordered mesoporous carbon–gold nanoparticle, OMC–GNPs, AuNPs and DNA tetrahedral nanostructures, TSPs), artificial enzymatic nanolabels ((Fe−P)_n_-MOF) and nanocarriers of signalling elements (Pt@PdNCs). The use of multifunctional hemin/G-quadruplex nanowires to simultaneously serve as bienzyme and direct electron mediator or MSNs as molecular gates have been exploited in other smartly designed configurations.

Fe_3_O_4_@pDA MNPs are appropriate solid nanosupports to develop an amperometric sandwich immunosensor for *L. pneumophila.* MIP-aptamer and MIP-DNA nanohybrids have been used as the basis for very attractive biosensing platforms for the determination of urea, TNT, heavy metals (Pb^2+^ and As^3+^) and 2,4-D. DNA-gated MSNs have been combined in an elegant approach with a PGM biosensor for Hg^2+^ determination. DNAzymes have been employed in combination with nanomaterials and other attractive amplification strategies, including CHA, the in situ DSDNA layer-by-layer assembly, the mimicking peroxidase activity of MnTMPyP and the use of TSPs as nanoscaffolds and RCA-formed multifunctional hemin/G-quadruplex nanowires, in the development of electrochemical biosensors for Pb^2+^.

It is noteworthy that although the reported methods show the great progress made in this field and the versatility and excellent capabilities that electrochemical affinity biosensors possess for the determination of relevant analytes in environmental and food analysis, there is still a long way to go for their integration into our daily lives. So far, these biosensors have been characterized and applied at research laboratory level, for single and rarely dual, determinations and with a limited number of samples (many of them spiked). In addition to make exhaustive validation studies with a larger number of samples and in the hands of different users and in different environments, the preparation of electrochemical biosensors on paper substrates and wearable formats by exploiting the screen-printed technology, will provide an indisputable attraction to these biosensors that will facilitate their acceptance in our society as a whole.

Considering the wide variety of bioreceptors, nanomaterials, bioassay formats and amplification strategies, it is clear that we face with a practically unlimited number of possible combinations to design biosensors on demand with compatible and tunable characteristics for the demanded application.

However, further work is still needed to be able to provide more affordable, eco-friendly and stable, reusable and with antibiofouling properties biosensors with capabilities beyond sensitivity and selectivity. The development of electrodes and nanostructures prepared from nontoxic materials, ecofriendly solvents and reagents, and using methodologies to reduce both the sample size and the amount of waste products to minimize the environmental impact must be encouraged. For example, the development of self-calibrating (or calibration-free) devices and able to support continuous, real-time measurements of analytes in unprocessed and/or flowing samples, with minimum use of reagents and minimal intervention by the user, would make biosensors even more attractive and suitable to fit into the busy real world. Indeed, although important steps have begun to be taken in these directions with promising progress, the coupling of novel nanomaterials with biomolecular switches (DNAs, aptamers or peptides that reversibly change conformation in response to the specific binding of a wide range of molecular targets) is particularly appealing to achieve the “ideal device”. Exciting routes to move forward include also a deeper exploitation of the unique properties of electrochemical biosensors for multiplexed determinations by coupling them more to portable, low-cost and custom-made instruments. All these achievements will boost their translational progress beyond the well-controlled laboratory benchtop and make electrochemical biosensors gain ground on other cumbersome methodologies to provide quality control and safety monitoring by nonspecialized users at the point of attention and even in resource limited settings.

Moreover, additional efforts should be made on the rational exploration of new nanomaterials and/or combinations thereof to improve the catalytic efficiency, selectivity, reproducibility and stability of the resulting biosensors, achieving the synergistic properties and functionalities. In addition, the modification of nanomaterials by adopting standardized, bioinspired and/or “safe-by-design” approaches for their synthesis and/or their coating will play a significant role in constructing biotools easily adopted by our increasingly demanding society.

In summary, although there are still several bottlenecks to overcome and we must be aware that we are still not close to their widespread commercialization and availability, the intense work performed and great interest aroused prove that it is a living field, tremendously active and that electrochemical affinity biosensors will bring us great surprises and unexpected possibilities in the coming years.

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
