# Peer review of "Electrochemical Affinity Biosensors Based on Selected Nanostructures for Food and Environmental Monitoring"

_sensors, 2020, doi:10.3390/s20185125_

Round 1

Reviewer 1 Report

The review attempted to summarize advances in nanostructures applied in sensor devices aimed at food or environmental samples. The subject area is of interest given the continuing interest in sensor devices in the environment and food sectors. The review provides an overview of recent applications in the area although the structure and level of detail needs to be improved. It is not apparent what “Rising Star Nanostructures” actually refers to given the nanostructures described are established technologies. The review was primarily application of nanostructures and transduction approaches. In this respect, the authors should consider restructuring the review in terms of nanostructure type. In addition, in many instances the authors simply referred to a sensor although gave no indication of performance. Of more concern is there was no indication on the advances made in nanostructures.

Specific points

The Abstract should be made more informative with regards to identifying advances relating to nanomaterials.

Title: Monitoring of what?

Line 14: What context?

Line 19: What problems?

Line 25: Which targets?

Line 29: Regulatory or AOAC approval is a further barrier.

Line 37: Which analytes and why has the need for testing increased?

Line 40: Alternative to what?

Line 43 and throughout: Statements should be supported by references.

Line 45: Need to describe how nanostructures can improve sensor performance.

Line 52: The authors need to describe the types of nanostructures available.

Line 64: Why mention gold nanoparticles here? Mimic enzyme activity with regards to reaction rates or selectivity? Would gold be considered inert?

Line 74: Low recovery has been a consistent issue with nanoparticles.

Line 80: The structure of the review could be improved. Magnetite seems to be suddenly introduced and could be introduced earlier.

Line 108: The authors need to define what they mean by “enzyme-like”?

Line 129: Authors should avoid using subjective terms.

Line 132: One step process to what?

Line 140: Need to give examples to sensors and performance.

Line 147: What would be considered “highly selective”?

Line 155: More details of applications should be provided.

Line 176: Good to elaborate on some examples.

Line 188: CFU’s

Line 199: What is considered “very-sensitive”?

Line 216: What nanostructures?

Line 231: How was the orientation of aptamers assured during printing.

Line 237: The change in charge transfer resistance is represented in a Nyquist plot.

Line 252: Isent it that the hydrogen peroxide generates the increase current using DVP?

Line 298: The authors need to report on the performance of the sensor relative to standard methods.

Line 307: The authors should define MOFs.

Line 537: From the review it was unclear which were new nanostructures given these were based on previously applied approaches.

Line 541: The review provide application of nanostructures rather than innovations.

Author Response

Reviewer 1

The review attempted to summarize advances in nanostructures applied in sensor devices aimed at food or environmental samples. The subject area is of interest given the continuing interest in sensor devices in the environment and food sectors. The review provides an overview of recent applications in the area although the structure and level of detail needs to be improved. It is not apparent what “Rising Star Nanostructures” actually refers to given the nanostructures described are established technologies. The review was primarily application of nanostructures and transduction approaches. In this respect, the authors should consider restructuring the review in terms of nanostructure type. In addition, in many instances the authors simply referred to a sensor although gave no indication of performance. Of more concern is there was no indication on the advances made in nanostructures.

Many thanks to this reviewer for his/her work, for recognizing the interest of this manuscript and for recommending publication in this journal after addressing the useful suggestions pointed out as indicated below.

After thinking in detail about the best way to organize this review, we decided to discuss and group the approaches by application field regarding the title of the special issue for which it has been prepared "Advanced Biosensors for Food and Environmental Monitoring" and in order to avoid an excessive number of subsections, taking into account the variability of nanostructures and their use in the selected works. In our opinion, the most relevant aspects of the sensors performance are discussed in the text, and further supported by the two extensive and detailed summary Tables included. Moreover, we consider that the versatility of use, modification and demonstrated applicability of selected nanostructures in the framework of the electrochemical affinity sensors highlighted and discussed exhaustively through the text show clearly the advances made in the field and the unique opportunities they provide.

Specific points

The Abstract should be made more informative with regards to identifying advances relating to nanomaterials.

Thank you very much for this comment. We have modified the abstract to make it more informative.

Title: Monitoring of what?

Evidently, monitoring of relevant analytes in these two fields. We think that the title of a review such as this should be left open as well as that of the Special issue in which is going to be included "Advanced Biosensors for Food and Environmental Monitoring".

Line 14: What context? Line 19: What problems? Line 25: Which targets?

The Abstract has been modified considering all the reviewer's suggestions to make it more informative.

Line 29: Regulatory or AOAC approval is a further barrier.

All the specific points listed below have been addressed in the revised manuscript unless otherwise specified.  

Line 37: Which analytes and why has the need for testing increased? Line 40: Alternative to what?

Line 43 and throughout: Statements should be supported by references. Line 45: Need to describe how nanostructures can improve sensor performance.

The Introduction section has also been revised addressing all the requested comments.

Line 52: The authors need to describe the types of nanostructures available.

This part has also been rewritten, indicating the nanostructures we have focused on in this review article, also including the new general Scheme 1 (in line also with the request of Reviewer 2).

Line 64: Why mention gold nanoparticles here? Mimic enzyme activity with regards to reaction rates or selectivity? Would gold be considered inert?

We mention the AuNPs here because they are one of the most commonly used nanomaterials in the preparation of individual or hybrid nanostructures. We have revised and expanded the paragraph regarding their most important properties and applications in the highlighted approaches. 

Line 74: Low recovery has been a consistent issue with nanoparticles.

This part of the text was also modified by pointing out the main drawbacks of MNPs.

Line 80: The structure of the review could be improved. Magnetite seems to be suddenly introduced and could be introduced earlier.

We couldn't think of a better way to structure this section and, in our opinion, magnetite is in the right place.

Line 108: The authors need to define what they mean by “enzyme-like”?

We have written: "... with inherent activity similar to enzymes are gaining ..." to avoid confusion.

Line 129: Authors should avoid using subjective terms.

Thank you for this comment. The adjectives "exquisite" and "extraordinary" have been deleted.

Line 132: One step process to what?

For more clarity, this sentence has been modified as follows: "… assembled on gold surfaces in a one-step through…"

Line 140: Need to give examples to sensors and performance.

In our opinion going deeper into this is both out of place in this section and out of scope for this review.

Line 147: What would be considered “highly selective”?

For more clarity, we have rewritten this sentence as: "…construction of highly selective sensors, able to determine the target compound in the presence of other similar species in complex samples, by means …"

Line 155: More details of applications should be provided.

We have completed this part as follows: "Specific bio-modifiers (aptamers and DNAs) for the target compounds have been recently used in combination with MIPs nanostructures to increase their selectivity and sensitivity and develop affinity biosensors for pesticides [56,57]."

Line 176: Good to elaborate on some examples.

We have exemplified this by including the corresponding references in Table 1 as follows: “…A limited number of biosensors uses unmodified electrode substrates and nanomaterials as nanocarriers and/or catalytic labels [32,71,74]…”

Line 188: CFU’s

Thank you for this comment. This acronym has been defined.

Line 199: What is considered “very-sensitive”?

We consider that affinity biosensors offering LODs values of pg mL for protein and aM for genetic biomarkers can be perfectly qualified as very sensitive.

Line 216: What nanostructures?

In order to provide more information, we have rewritten this paragraph as follows: "… nanostrured with nanocomposites of reduced graphene oxide and AuNPs (AuNPs-rGO, Figure 2) [75] or graphene oxide (GO) [76]…"

Line 231: How was the orientation of aptamers assured during printing.

Unfortunately, nothing is mentioned in the article related to the aptamer orientation and so, this has been indicated in the revised manuscript as follows: "…high-resolution patternability (there is no discussion about the orientation of the aptamer) for the impedimetric…"

Line 237: The change in charge transfer resistance is represented in a Nyquist plot.

We agree with the reviewer and, accordingly, we have written: "…charge transfer resistance, Rct, corresponding to the semicircle diameter in the Nyquist plot…"

Line 252: Isent it that the hydrogen peroxide generates the increase current using DVP?

This question has been clarified by writing the following sentence: "… by the displacement of the aptamer adsorbed on their surface and corresponding to the reduction of the oxidized Thi generated in the reduction of H2O2 by AuNPs."

Line 298: The authors need to report on the performance of the sensor relative to standard methods.

To clarify this issue, the following text has been added: "…transfer efficiency. This strategy combining aptasensing and molecular imprinting allowed designing a highly sensitive method. A…"

Line 307: The authors should define MOFs.

Thank you for this comment but MOFs was previously defined in line 169.

Line 537: From the review it was unclear which were new nanostructures given these

were based on previously applied approaches.

We agree with the Reviewer and to avoid confusion we have rewritten the beginning of section 5 as follows: "Undoubtedly, the use of nanostructures as electrode modifiers and/or specialized labels has been has been responsible for the excellent capabilities…"

Line 541: The review provide application of nanostructures rather than innovations.

In the second paragraph of section 5, electrochemical affinity biosensors are discussed in general and it is unquestionable that innovations in the modification and use of the highlighted nanostructures are directly responsible for the attractive operation and the reported applications. This important aspect has been clarified in section 5.

Reviewer 2 Report

In this review manuscript, the authors summarized the outstanding properties and applications of electrochemical affinity biosensors offered by selected nanostructures-based electrochemical affinity biosensors involving selected nanostructures for agri-food and environmental applications reported in the last 5 years. There are four main sections in the manuscript. At first, they listed the Rising Star Nanostructures used for the preparation of Electrochemical Affinity Biosensors, and stated the advantages of these materials. Then, the applications of electrochemical affinity biosensors based on these Rising Star Nanostructures in the agri-food and environmental fields were reviewed in the following two sections. Finally, the challenges and future perspectives are further discussed. The manuscript is well written. Therefore, I suggest it is acceptable after the following minor modifications.

  1. Section 1. Introduction. There are only four lines for the introduction of the background of this work. This section should be more informative. For example, the definition, the development, and the advantages of the electrochemical affinity biosensors might be introduced. Some related review papers are provided for reference, TrAC Trends Anal. Chem. 129 (2020) 115943, Nanoscale 11(2019) 19105.
  2. Section 2 Rising Stars Nanostructures in Electrochemical Affinity Biosensing. The reason to choose these Rising Stars Nanostructures to reviewed in this work should be discussed.
  3. One or two summary figures of the whole work or the nanomaterials discussed here should be offered in section 1 or 2.
  4. Section 3 and section 4. Although the content is wide-ranging and technical details are described, there should be a simple introduction at the beginning of each section, which might state the classification basis of the following illustrated examples, or the reasons for choosing the following illustrated examples to discuss.
  5. Some related studies published more recently that provided different electrochemical biosensing methods can be included in sections 3 or 4, for example, Food chemistry 271(2019) 54, Sensor Actuat B-Chem. 283 (2019) 262, Microchim Acta 187, 442 (2020).
  6. Section 5 General Considerations, Challenges, and Perspectives. Although the several sentences mentioning the challenges and perspectives of electrochemical affinity biosensors, such as “there is still a long way to go for their integration into our daily lives”, “further work is still needed to be able to provide more affordable, eco-friendly…”, a detailed discussion of challenges and perspectives might be concluded more deeply.
  7. As stated in section 5, “the development of self-calibrating (or calibration-free) devices and able to support continuous, real-time measurements of analytes…” is one of the future highlighted works, some published works about such studies are suggested to be reviewed in sections 3 or 4, for example, Anal. Chem. 90(2018) 10641, Biosens. Bioelectron 152(2020) 112010.

Author Response

Reviewer 2

In this review manuscript, the authors summarized the outstanding properties and applications of electrochemical affinity biosensors offered by selected nanostructures-based electrochemical affinity biosensors involving selected nanostructures for agri-food and environmental applications reported in the last 5 years. There are four main sections in the manuscript. At first, they listed the Rising Star Nanostructures used for the preparation of Electrochemical Affinity Biosensors, and stated the advantages of these materials. Then, the applications of electrochemical affinity biosensors based on these Rising Star Nanostructures in the agri-food and environmental fields were reviewed in the following two sections. Finally, the challenges and future perspectives are further discussed. The manuscript is well written. Therefore, I suggest it is acceptable after the following minor modifications.

Thank you very much for recognizing the quality of this manuscript and for recommending its acceptance after addressing the useful suggested modifications.

Section 1. Introduction. There are only four lines for the introduction of the background of this work. This section should be more informative. For example, the definition, the development, and the advantages of the electrochemical affinity biosensors might be introduced. Some related review papers are provided for reference, TrAC Trends Anal. Chem. 129 (2020) 115943, Nanoscale 11(2019) 19105.

Thank you very much for this comment and pointing out these interesting references. We have expanded section 1 by including relevant information and additional references to make it more informative.

Section 2 Rising Stars Nanostructures in Electrochemical Affinity Biosensing. The reason to choose these Rising Stars Nanostructures to reviewed in this work should be discussed.

One or two summary figures of the whole work or the nanomaterials discussed here should be offered in section 1 or 2.

We have selected these target nanostructures because they have only recently used in electrochemical affinity biosensor configurations. In addition, as will become apparent throughout this review article, the great versatility of modification and use they provide and the unique attributes they impart to the resulting biosensors which offer very attractive performance characteristics and potential for addressing pioneering applications in the agro-food and environmental areas allow predicting their interest and use will continue to grow in the coming years.

We have clarified this point by adding this text at the end of section 1 “…These nanostructures have been selected because they have only recently used in electrochemical affinity biosensor configurations. In addition, the great versatility of modification and use they provide and the unique attributes they impart to the resulting biosensors allowing pioneering applications in the agro-food and environmental areas, lead to predict an increasing interest for their use in the coming years.”

The requested schematic display of the review concept summarizing the fundamental of nanostructures-based electrochemical affinity biosensors is given in new Scheme 1.

Section 3 and section 4. Although the content is wide-ranging and technical details are described, there should be a simple introduction at the beginning of each section, which might state the classification basis of the following illustrated examples, or the reasons for choosing the following illustrated examples to discuss.

Thank you very much for this suggestion. We have included a small introductory paragraph at the beginning of each of these sections about the approaches to be discussed, points to pay attention to and the criteria followed for their classification.

Some related studies published more recently that provided different electrochemical biosensing methods can be included in sections 3 or 4, for example, Food chemistry 271(2019) 54, Sensor Actuat B-Chem. 283 (2019) 262, Microchim Acta 187, 442 (2020).

Thank you very much for writing down these interesting references. Please note that reference Sensor Actuat B-Chem. 283 (2019) 262 was already included in the first submission (reference 17 and 21 in the original and revised version, respectively). Reference Food chemistry 271 (2019) 54 reports an electrochemical sensor using double stranded DNA (dsDNA)/Hemoglobin (Hb)-modified screen printed gold electrode for the determination of acrylamide in potato fries but the method did not involve the use of nanomaterials and, therefore, we consider it out of the main scope of this review. The manuscript Microchim Acta 187, 442 (2020), which overviews recent developments in synthesis and modification aspects of carbon-based two-dimensional materials for electrochemical sensors has been included at the beginning of section 2 (new reference [7]).  

Section 5 General Considerations, Challenges, and Perspectives. Although the several sentences mentioning the challenges and perspectives of electrochemical affinity biosensors, such as “there is still a long way to go for their integration into our daily lives”, “further work is still needed to be able to provide more affordable, eco-friendly…”, a detailed discussion of challenges and perspectives might be concluded more deeply.

Thank you very much for this comment, we have reviewed section 5 to discuss in more detail the issues pointed out.

As stated in section 5, “the development of self-calibrating (or calibration-free) devices and able to support continuous, real-time measurements of analytes…” is one of the future highlighted works, some published works about such studies are suggested to be reviewed in sections 3 or 4, for example, Anal. Chem. 90(2018) 10641.

This comment is also appreciated. As far as we know (Sensors 2020, 20, 3376; doi:10.3390/s20123376) the electrochemical affinity biosensors exhibiting calibration-free features and supported continuous and real-time analyte measurements in food and environmental applications are mostly non-nanostructured switch-based biosensors. Regarding the published works kindly pointed out by the reviewer, the work reported in Anal. Chem. 90 (2018) 10641 does not involve the use of nanostructures and the work reported in reference Biosens. Bioelectron 152 (2020) 112010, apart from exploiting other types of nanostructures than those we have focused on, it was applied to clinical samples. With this in mind, only the method reported in this latter reference has been discussed in the revised manuscript (end of section 4, new reference 91).

Reviewer 3 Report

The manuscript entitled “Electrochemical Affinity Biosensors based on Rising Star Nanostructures for Food and Environmental Monitoring“ submitted by the group of Authors represents a comprehensive review on the topic. The Authors made a tremendous effort to summarize most of the current novelties and approaches in electrochemical affinity biosensors within the proposed topic. My compliments to the successful work.

There are two main points I would like to address:

  • I would suggest the Authors to include more referencing in the text where they describe the sensors, working principles, or results. This is missing very often in the manuscript. My point is to refer to certain references, and not to do the referencing without actual mentioning of the reference.
  • I would like to stress out an interesting paper recently published in Sensors202020(1), 274; https://doi.org/10.3390/s20010274 that was not mentioned in the review. It deals with affinity detection of E. coli by EIS. I recommend the Authors to include the principles and strategies in the manuscript.

For this reasons, I would recommend the minor revision of the manuscript.

Author Response

Reviewer 3

The manuscript entitled “Electrochemical Affinity Biosensors based on Rising Star Nanostructures for Food and Environmental Monitoring” submitted by the group of Authors represents a comprehensive review on the topic. The Authors made a tremendous effort to summarize most of the current novelties and approaches in electrochemical affinity biosensors within the proposed topic. My compliments to the successful work.

We are very grateful to the referee for his/her work, for recognizing the efforts dedicated to its preparation and for the encouraging comments and recommending its publication after minor revision. 

There are two main points I would like to address:

I would suggest the Authors to include more referencing in the text where they describe the sensors, working principles, or results. This is missing very often in the manuscript. My point is to refer to certain references, and not to do the referencing without actual mentioning of the reference.

We have revised the text to support it with more references.

I would like to stress out an interesting paper recently published in Sensors 2020, 20(1), 274; https://doi.org/10.3390/s20010274 that was not mentioned in the review. It deals with affinity detection of E. coli by EIS. I recommend the Authors to include the principles and strategies in the manuscript.

Thank you very much for writing down this interesting paper. Although we fully agree with the interest and practical applicability of the biosensor it reports, it does not exploit the use of nanomaterials so that we consider it is outside the main core of this review.

For this reasons, I would recommend the minor revision of the manuscript.

We feel that these additional changes have improved the manuscript even more. We are confident that the current version complies with the high standards and requirements of Sensors, and is now suitable for publication in this Journal.

Thank you for your attention and for the very useful suggestions.

Please do not hesitate to contact us if you need additional information.

We thank you in advance for your attention!

Sincerely,

Dr. Susana Campuzano

[email protected]

Prof. Paloma Yañez-Sedeño                                                             

[email protected]

Round 2

Reviewer 1 Report

The authors have addressed some of the issues raised in the original review although the inherent weaknesses persist. The title still refers to “Rising Star” but the authors still failed to define what the term means in the context of the review. The reality is that the origins of the nanostructures referred to in the review have been in development for 20 years. In this regard, the review is more related to application in sensor devices rather than innovations in sensor technologies. The sub-division between food and environmental applications would lend itself to splitting the sections into analyte focus rather than recognition-molecule type. For example, sub-headings referring to sensing heavy metals would describe sensors for detecting the analyte. As it is, the text is repetitive and without the sub-headings, comes across as confusing. Consequently, the authors should consider re-organizing the script and refer more to application of nanostructures rather than suggest innovations in the area.

Author Response

The authors have addressed some of the issues raised in the original review although the inherent weaknesses persist.

We thank this referee for his/her work and comments. In fact, we believe that in the first revision, we took into consideration all his/her suggestions and made the requested changes. We regret that this reviewer still considers that "inherent weaknesses persist". Fortunately, this is not the opinion of the other two reviewers.

The title still refers to “Rising Star” but the authors still failed to define what the term means in the context of the review. The reality is that the origins of the nanostructures referred to in the review have been in development for 20 years.

During R1 we pointed out at the end of section 1 what the term "Rising Star" meant in the context of the review "These nanostructures have been selected because they have been only recently used in electrochemical affinity biosensor configurations. In addition, the great versatility of modification and use they provide and the unique attributes they impart to the resulting biosensors allowing pioneering applications in the agro-food and environmental areas, lead to predict an increasing interest for their use in the coming years." Nevertheless, we agree with the reviewer that these nanostructures are not new. Conversely, their use for the preparation of electrochemical affinity biosensors, the subject of the review article, is novel and that is what we have pointed out. However, to avoid misunderstandings we have changed through the text "Rising Star" by "Selected".

In this regard, the review is more related to application in sensor devices rather than innovations in sensor technologies.

While we respect this subjective view of the referee, we never said that this article covered "innovations in sensor technologies". We are sure that the readers will not expect this from the title of the article: "Electrochemical Affinity Biosensors based on Selected Nanostructures for Food and Environmental Monitoring". This is stressed in the abstract "This review article aims to give, by highlighting representative methods reported in the last 5 years, an updated and general overview of the main improvements that the use of such well-ordered nanomaterials as electrode modifiers or advanced labels confer to electrochemical affinity biosensors in terms of sensitivity, selectivity, stability, conductivity and biocompatibility focused on food and environmental applications, less covered in the literature than clinics.".

Indeed, if we had wanted to overview "innovations in sensor technologies", we would have written a totally different manuscript.

However, we should recognize that though it was not our purpose, when describing the basis of each highlighted biosensor, the innovations of these biodevices in terms of manufacture, modification and use of particular nanostructures, which are largely responsible for the attractive features and applications, are implicit. This point was also clarified during R1 in the second paragraph of section 5 "Innovations in the manufacture, modification and use of particular nanostructures are largely responsible for the attractive features and applications shown by the selected biosensors".

The sub-division between food and environmental applications would lend itself to splitting the sections into analyte focus rather than recognition-molecule type. For example, sub-headings referring to sensing heavy metals would describe sensors for detecting the analyte.

We consider it more appropriate to classify the selected representative methods according to the type of affinity biosensor and field of application than to the target analyte, particularly in a review focused on electrochemical affinity biosensors to be included in the Special Issue "Advanced Biosensors for Food and Environmental Monitoring: A Themed Issue Dedicated to Professor Jean-Louis Marty".

We would like to note that a classification regarding the target analyte would lead to a high number of subsections (at least ten, devoted to food allergens, gluten, GMOs, toxins, antibiotics, pesticide residues, bacteria, toxins, fungus, yeast and oestrogens). In addition, there are analytes such as bacteria and their toxins that have been determined both in the food and environmental fields. Therefore, we think that an analyte-related classification would make the main focus of the review and of the special issue “Advanced Biosensors for Food and Environmental Monitoring”, to be lost. In our opinion, this review allows raising awareness of the broad scope of applications of electrochemical affinity biosensors in particularly important areas of food and environmental analysis.

As it is, the text is repetitive and without the sub-headings, comes across as confusing.

In our view, the classification made, besides being more logical, gives homogeneity to the review and does not make the text repetitive or confusing (nor in the opinion of the other two referees who judged the article).

Consequently, the authors should consider re-organizing the script and refer more to application of nanostructures rather than suggest innovations in the area.

We are sorry but we find this comment contradictory to what this referee has previously said. It seems that he/she is now inviting us to refer more to applications than to innovations when he/she has previously said the opposite. We can't emphasize enough that this article aims to give an updated overview of the opportunities offered by electrochemical affinity biosensors in the environmental and food fields by the ingenious use of nanostructures in their manufacture.